# Domain Indexing Collaborative Filtering for Recommender Systems

## Abstract

In cross-domain recommendation systems, addressing cold-start items remains a significant challenge. Previous methods typically focus on maximizing performance using cross-domain knowledge, often treating the knowledge transfer process as a black box. However, the recent development of domain indexing introduces a new approach to better address such challenges. We have developed an adversarial Bayesian framework, Domain Indexing Collaborative Filtering (DICF), that infers domain indices during cross-domain recommendation. This framework not only significantly improves the recommendation performance but also provides interpretability for cross-domain knowledge transfer. This is verified by our empirical results on both synthetic and real-world datasets.

## 1 Introduction

In recommender systems, prior user-item interactions are crucial for facilitating accurate recommendations. However, when an item has not previously interacted with any users – known as a "cold-start" item – it becomes challenging to provide high-quality recommendations. This issue is common in cross-domain recommendation, where we train the recommender system on user-item interactions from a source domain but, during inference, encounter items from other domains that the system has never seen before. For example, a recommender system is trained on users and items from the United States, but during inference, it needs to handle items from the United Kingdom.

Extensive efforts have been made to address this problem. For instance, Jiang et al. (2016) proposed a semi-supervised transfer learning approach that uses overlapped-user-based similarities to regularize matrix factorization results. Zhu et al. (2019) introduced special embedding layers to create unique embeddings for users and items in each domain, while the embeddings of overlapping users are a combination of embeddings from different domains. Wang et al. (2015) incorporated additional item and user content information to generate more robust latent representations across domains. While these methods leverage cross-domain information to enhance performance, they often treat the transfer process as a "black box," limiting our understanding and hindering further model improvements. This issue is critical because it can obscure the model's decision-making process, making it difficult to trust and audit.

Recent advances in Domain Adaptation, particularly in domain indexing (Wang et al., 2020; Liu et al., 2023; Xu et al., 2023; 2022), offer a new perspective on this problem. A domain index – either a scalar or a vector that encodes domain semantics – has been shown to improve model generalization while providing insights into the transfer process.

Inspired by this idea, we propose an adversarial Bayesian framework, dubbed *Domain Indexing Collaborative Filtering (DICF)*, which infers domain indices during cross-domain recommendation. The core idea is to aggregate and distill domain-specific features, which then serve as the domain index. This domain index, combined with adversarial learning, enables the model to learn generalizable features, explore domain relationships, and highlight spurious information during cross-domain recommendation, thereby improving our understanding of the process.

**Example 1.** *Examples to Illustrate Domain Indices for Recommendation and Interpretability]*
*Consider the tissue products from Japan, Germany, and Spain, which, despite some similarities, vary significantly in descriptions due to language differences. If a recommendation model is trained only using data from Japan and Germany (source domains), it may perform poorly in the Spain*

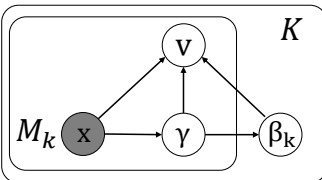

Figure 1: **Left:** DICF's generative model. The dash line between $\boldsymbol{\beta}_k$ and $\mathbf{v}$ indicates that we enforces the independence between $\boldsymbol{\beta}_k$ and $\mathbf{v}$. **Right:** DICF's inference model. Variables with grey backgrounds represent observed variables, while all other variables are latent.

*(target domain) due to these linguistic discrepancies. To mitigate this, "unnecessary" or spurious features such as the language of product descriptions across domains should be removed. This resembles standardizing descriptions to English for all domains. A latent variable, termed a domain index, is learned to facilitate this removal. For example, a domain index might be an embedding that denotes the language (e.g., German, Japanese, Spanish) used in product descriptions. This index helps indicate how to eliminate spurious features and effectively serves as a "domain embedding" that captures the essence of the domain. For instance, the domain index for German tissues might be closer to that of Spanish tissues than to Japanese tissues. Such a domain index could improve both the generalization and the interpretability of the model. Further discussion on this will be in Sec. 2.2, "Model Intuition."*

In summary, our contributions are as follows:

- We introduce a novel adversarial Bayesian method, Domain Indexing Collaborative Filtering (DICF), for inferring domain indices in cross-domain recommendation.
- Our experiments on both synthetic and real-world datasets demonstrate that DICF significantly outperforms state-of-the-art methods.
- Visualizations of the domain indices learned by DICF reveal insights into the transfer process, enhancing the interpretability of cross-domain recommendations.

## 2 METHOD

In this section, we provide a brief overview of our method, covering the problem setting, model intuition, probabilistic graphical model, and objective function.

### 2.1 PROBLEM SETTING AND NOTATION

In this paper, we address recommendation problems involving a total number of $N$ users and $M$ items, with the items divided into $K$ domains, and each domain $k$ containing $M_k$ items. Each item is characterized by content features represented as an $M \times J$ matrix $\mathbf{x}$, while the user content is given by an $N \times J$ matrix $\mathbf{u}$. The user-item interactions are represented by an $N \times M$ binary matrix $R$, where a value of "1" indicates that the user likes the item, while "0" indicates either a lack of preference or no interaction with the item. Our model is trained on interactions (ratings) between users and items across $K_s$ source domains, aiming to predict interactions for items in $K_t$ target domains for all users, where $K_s + K_t = K$. Notably, all items in the target domains are cold-start items, meaning that they are not present in the training set.

### 2.2 MODEL INTUITION

In this subsection, we explain what we mean by the "domain index" in the context of cross-market recommendation, discuss how it facilitates model interpretation and performance, and describe how to derive such a domain index.

In product recommendations, auxiliary data often improves outcomes. For instance, each product typically includes product descriptions and tags – which we refer to as item contexts. Our goal is to learn item representations that generalize effectively from these contexts. Thus, we must eliminate

non-generalizable (spurious) features during feature embedding. For example, the same item may have contexts in different languages. While product information like brand and price is crucial, the language used should not affect recommendations. Therefore, we separate features into useful and spurious ones, using only the useful for prediction. Spurious features are domain-specific and cannot generalize across domains; thus, they uniquely identify each domain, making them a good source for the domain index.

The *domain index*, aggregated from these spurious features, captures relationships among domains. For instance, we might find that the domain index for U.S. products is closer to that of Mexico than to that of Japan, indicating how the model transfers across different markets. Moreover, since the domain indices embed these domain relationships, we include them as additional inputs to the model, further enhancing performance, as verified in our experiments.

To derive the domain index in an unsupervised manner, we leverage two properties of spurious features: (1) they do not generalize across domains, and (2) they do not facilitate recommendation. Accordingly, we perform this decomposition using adversarial learning. We infer two types of features: the domain index and domain-invariant features. We enforce independence between the domain index and domain-invariant features using a discriminator. We then use only the domain-invariant features for subsequent rating prediction. This ensures that prediction-related information is retained in domain-invariant features, while domain-specific information is captured in the domain index, as proved by Xu et al. (2023). Importantly, our domain index differs from general domain-dependent features: it is a domain-level variable shared by all data points in the same domain, not an instance-level variable. This enables the domain index to uniquely represent each domain.

### 2.3 PROBABILISTIC GRAPHICAL MODEL OF DICF

Based on the intuition above, we propose a hierarchical Bayesian deep learning model, Domain Indexing Collaborative Filtering (DICF), to achieve this goal. It follows the generative process illustrated in Fig. 1 (left).

**Generative Process of DICF.** We assume the generative process below for DICF:

- For each domain $k$, a domain index $\boldsymbol{\beta}_k$ is generated from a prior distribution $p_\theta(\boldsymbol{\beta}|\boldsymbol{\alpha})$.
- Using $\boldsymbol{\beta}_k$, we generate a domain-specific feature $\boldsymbol{\gamma}$ for each item within domain $k$ from $p_\theta(\boldsymbol{\gamma}|\boldsymbol{\beta}_k)$.
- $\boldsymbol{\gamma}$ is then used to generate the item content features $\mathbf{x}$ from $p_\theta(\mathbf{x}|\boldsymbol{\gamma})$, where $\mathbf{x}$ represents the item's observable attributes.
- The item latent vector $\mathbf{v}$ is generated from distribution $p_\theta(\mathbf{v}|\boldsymbol{\beta}_k, \boldsymbol{\gamma}, \mathbf{x})$ .
- Finally, the predicted rating $\mathbf{R}$ for each user-item pair is computed using the user vector $\mathbf{u}$ and the item latent vector $\mathbf{v}$ with distribution $p_\theta(\mathbf{R}|\mathbf{u}, \mathbf{v})$. Note that the user vector is generated from the user context using pretrained encoders and is treated as an observed variable.

**Inference Process of DICF.** In the inference process, we aim to estimate the latent variables $\boldsymbol{\gamma}$, $\boldsymbol{\beta}_k$, and $\mathbf{v}$ from the observed data, as illustrated in Fig. 1 (right). The steps are as follows:

- Given the observed item content features $\mathbf{x}$, we infer the local domain-specific feature $\boldsymbol{\gamma}$ using $q_\phi(\boldsymbol{\gamma}|\mathbf{x})$.
- We aggregate the inferred $\boldsymbol{\gamma}$ values for all items within domain $k$ to estimate the domain indices $\boldsymbol{\beta}_k$ with distribution $q_\phi(\boldsymbol{\beta}_k|\boldsymbol{\gamma})$, capturing domain-level patterns.
- We infer the item embedding vectors $\mathbf{v}$ based on the estimated $\boldsymbol{\beta}_k$, the content features $\mathbf{x}$, and the local item indices $\boldsymbol{\gamma}$ with distribution $q_\phi(\mathbf{v}|\boldsymbol{\beta}_k, \boldsymbol{\gamma}, \mathbf{x})$.

**Model Factorization.** As shown in Fig. 1 (left), we factorize the generative model $p_\theta(\mathbf{x}, \mathbf{u}, \mathbf{v}, \boldsymbol{\beta}, \boldsymbol{\gamma}, \mathbf{R}|\boldsymbol{\alpha})$ into five conditional distributions:

$$p_\theta(\mathbf{x}, \mathbf{u}, \mathbf{v}, \boldsymbol{\beta}, \boldsymbol{\gamma}, \mathbf{R}|\boldsymbol{\alpha}) = p_\theta(\mathbf{R}|\mathbf{u}, \mathbf{v})p_\theta(\mathbf{v}|\mathbf{x}, \boldsymbol{\gamma}, \boldsymbol{\beta})p_\theta(\mathbf{x}|\boldsymbol{\gamma})p_\theta(\boldsymbol{\gamma}|\boldsymbol{\beta})p_\theta(\boldsymbol{\beta}|\alpha). \tag{1}$$

Here, $\boldsymbol{\theta}$ represents the set of parameters for the generative model. We assume that all five distributions follow a Gaussian distribution:

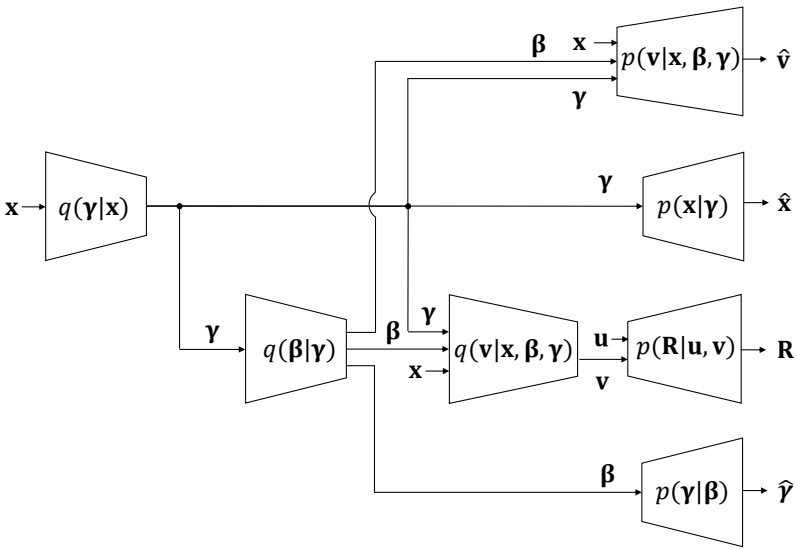

Figure 2: Network structure. For simplicity, we omit the subscripts of $q_\phi$ and $p_\theta$. Intuitively, the functions $q(\gamma|\mathbf{x})$, $q(\beta|\gamma)$, $q(\mathbf{v}|\mathbf{x},\beta,\gamma)$ and $p(\mathbf{R}|\mathbf{u},\mathbf{v})$ generate the instance-level domain-specific feature $\gamma$, the domain index $\beta$, the item vector $\mathbf{v}$, and the rating $\mathbf{R}$, respectively. Meanwhile, $p(\gamma|\mathbf{x})$, $p(\beta|\gamma)$ and $p(\mathbf{v}|\mathbf{x},\beta,\gamma)$ are used to produce reconstructed $\hat{\mathbf{x}}$, $\hat{\beta}$ and $\hat{\mathbf{v}}$.

$$p_\theta(\beta|\alpha) = \mathcal{N}(\boldsymbol{\mu}_\alpha, \boldsymbol{\sigma}_\alpha^2), \tag{2}$$

$$p_\theta(\gamma|\beta) = \mathcal{N}(\mu_\gamma(\beta;\theta), \sigma_\gamma^2(\beta;\theta)), \tag{3}$$

$$p_\theta(\mathbf{x}|\gamma) = \mathcal{N}(\mu_x(\gamma;\theta), \sigma_x^2(\gamma;\theta)), \tag{4}$$

$$p_\theta(\mathbf{v}|\mathbf{x},\gamma,\beta) = \mathcal{N}(\mu_v(\mathbf{x},\gamma,\beta;\theta), \sigma_v^2(\mathbf{x},\gamma,\beta;\theta)), \tag{5}$$

$$p_\theta(\mathbf{R}_{ij}|\mathbf{u}_i,\mathbf{v}_j) = \mathcal{N}(\mathbf{u}_i^T\mathbf{v}_j, \sigma_{\mathbf{R}_{ij}}^2\mathbf{I}), \tag{6}$$

where in Eqn. 6, $\mathbf{R}_{ij}$ represents the rating given by user $i$ to item $j$, while $\mathbf{u}_i$ and $\mathbf{v}_j$ denote the feature embeddings for user $i$ and item $j$, respectively. In our model, $\mathbf{R}_{ij}$ is a binary variable that can take values 0 or 1. We set the variance $\sigma_{\mathbf{R}_{ij}}^2 = \frac{1}{a}$ when $\mathbf{R}_{ij} = 0$ and $\sigma_{\mathbf{R}_{ij}}^2 = \frac{1}{b}$ when $\mathbf{R}_{ij} = 1$. To address the sparsity issue, we choose $a \ll b$.

To approximate the posterior distributions of the latent variables $p_\theta(\mathbf{v},\beta,\gamma|\mathbf{x})$, we employ an inference distribution $q_\phi(\mathbf{v},\beta,\gamma|\mathbf{x})$. As illustrated in Fig. 1 (right), we decompose $q_\phi(\mathbf{v},\beta,\gamma|\mathbf{x})$ as

$$q_\phi(\mathbf{v},\beta,\gamma|\mathbf{x}) = q_\phi(\gamma|\mathbf{x})q_\phi(\beta|\gamma)q_\phi(\mathbf{v}|\mathbf{x},\gamma,\beta), \tag{7}$$

where $\phi$ represents the set of parameters for the inference model. More specifically, we have:

$$q_\phi(\gamma|\mathbf{x}) = \mathcal{N}(\mu_\gamma(\mathbf{x};\phi), \sigma_\gamma^2(\mathbf{x};\phi)), \tag{8}$$

$$q_\phi(\beta|\gamma) = \mathcal{N}(\mu_\beta(\gamma;\phi), \sigma_\beta^2(\gamma;\phi)), \tag{9}$$

$$q_\phi(\mathbf{v}|\mathbf{x},\gamma,\beta) = \mathcal{N}(\mu_v(\mathbf{x},\gamma,\beta;\phi), \sigma_v^2(\mathbf{x},\gamma,\beta;\phi)). \tag{10}$$

Note that $\mu_.(\cdot;\cdot)$ and $\sigma_.(\cdot;\cdot)$ denote neural networks; $\theta$, $\phi$ are neural network parameters. The full network structure is illustrated in Fig. 2, where each neural network estimates the density function of each corresponding distribution.

We highlight several key insights for this network structure:

- **Encoder-Decoder Structure for** $q_\phi(\gamma|x)$ **and** $p_\theta(\mathbf{x}|\gamma)$. $p_\theta(\mathbf{x}|\gamma)$ aims to reconstruct $\mathbf{x}$ given $\gamma$, encouraging $\gamma$ to preserve as much item context information as possible.

- **Encoder-Decoder Structure for** $q_\phi(\beta|\gamma)$ **and** $p_\theta(\gamma|\beta)$. Here, $q_\phi(\beta|\gamma)$ aggregates the domain-specific features $\gamma$ to generate the domain index $\beta$. We sample the reconstructed $\hat{\gamma}$ from $p_\theta(\gamma|\beta)$ to ensure that $\beta$ captures comprehensive information from $\gamma$.

- $p_\theta(\mathbf{v}|\mathbf{x},\beta,\gamma)$ **regularizes** $q_\phi(\mathbf{v}|\mathbf{x},\beta,\gamma)$. During training, while $q_\phi(\mathbf{v}|\mathbf{x},\beta,\gamma)$ generates the current latent item vector $\mathbf{v}$, $p_\theta(\mathbf{v}|\mathbf{x},\beta,\gamma)$ produces another vector $\hat{\mathbf{v}}$ that remains close to the latent vector $\mathbf{v}$ from the previous epoch. This $\hat{\mathbf{v}}$ serves to constrain $\mathbf{v}$, preventing it from deviating significantly from the previous epoch's result, thereby functioning as a regularizer.

- $p_\theta(\mathbf{R}|\mathbf{u}, \mathbf{v})$ **predicts ratings,** similar to the general approach used in matrix factorization.

We follow the approach of Xu et al. (2023) in handling $q_\phi(\boldsymbol{\beta}|\boldsymbol{\gamma})$: First, we aggregate all domain-specific features, $\boldsymbol{\gamma}$, for each domain. Next, we compute a domain distance matrix by measuring the Earth Mover's distance between each set of features. This matrix is then decomposed using multi-dimensional scaling (MDS) to obtain domain embeddings. These embeddings are subsequently fed into a neural network to generate the final domain index.

## 2.4 Loss Function

**Evidence Lower Bound (ELBO).** To train the generative and inference models, we employ the evidence lower bound (ELBO) as our objective. By maximizing the ELBO, we can learn the optimal variational distribution $q_\phi(\mathbf{v}, \boldsymbol{\beta}, \mathbf{z}|\mathbf{x})$, which serves as the best approximation to the true posterior distribution of the latent variables $p_\theta(\mathbf{v}, \boldsymbol{\beta}, \boldsymbol{\gamma}|\mathbf{x})$. The ELBO is given by:

$$\mathcal{L}_{ELBO}(\mathbf{x}, y) = \mathbb{E}_{q_\phi(\mathbf{v}, \boldsymbol{\beta}, \boldsymbol{\gamma}|\mathbf{x})}[\log p_\theta(\mathbf{x}, \mathbf{u}, \mathbf{v}, \boldsymbol{\beta}, \gamma, \mathbf{R}|\boldsymbol{\alpha})] - \mathbb{E}_{q_\phi(\mathbf{v}, \boldsymbol{\beta}, \boldsymbol{\gamma}|\mathbf{x})}[\log q_\phi(\mathbf{v}, \boldsymbol{\beta}, \boldsymbol{\gamma}|\mathbf{x})]. \tag{11}$$

With the factorization in Eqn. 1 and Eqn. 7, we decompose the ELBO as (omitting $\boldsymbol{\alpha}$ to avoid clutter):

$$\mathcal{L}_{ELBO}(\mathbf{x}, y) = \mathbb{E}_{q_\phi(\boldsymbol{\gamma}|\mathbf{x})}[\log p_\theta(\mathbf{x}|\boldsymbol{\gamma})] \tag{12}$$

$$+ \mathbb{E}_{q_\phi(\mathbf{v}, \boldsymbol{\beta}, \boldsymbol{\gamma}|\mathbf{x})}[\log p_\theta(R|\mathbf{u}, \mathbf{v})] \tag{13}$$

$$+ \mathbb{E}_{q_\phi(\boldsymbol{\gamma}|\mathbf{x})}\mathbb{E}_{q_\phi(\boldsymbol{\beta}|\boldsymbol{\gamma})}[\log p_\theta(\boldsymbol{\gamma}|\boldsymbol{\beta})] \tag{14}$$

$$- \mathbb{E}_{q_\phi(\mathbf{v}, \boldsymbol{\gamma}, \boldsymbol{\beta}|\mathbf{x})}\big[KL[q_\phi(\beta|\boldsymbol{\gamma})||p_\theta(\boldsymbol{\beta})]\big] - KL[q_\phi(\mathbf{v}|\mathbf{x}, \boldsymbol{\gamma}, \boldsymbol{\beta})||p_\theta(\mathbf{v}|\mathbf{x}, \boldsymbol{\gamma}, \boldsymbol{\beta})] - \mathbb{E}_{q_\phi(\boldsymbol{\gamma}|\mathbf{x})}[\log q_\phi(\boldsymbol{\gamma}|\mathbf{x})], \tag{15}$$

network parameterization (see the network structure in Fig. 2); Here, Eqn. 12 and Eqn. 14 serve as the reconstruction loss, while Eqn. 13 performs rating regression. Eqn. 15 serves as the regularization term. Note that for target domain items, Eqn. 13 is excluded.

**Discriminator with an Adversarial Loss.** To ensure independence between $\boldsymbol{\beta}$ and $\mathbf{v}$, we introduce an additional discriminator $D$ that is trained using an adversarial loss while simultaneously maximizing the ELBO in Eqn. 11. As demonstrated in Xu et al. (2023), optimizing this adversarial loss to its optimal guarantees that $\boldsymbol{\beta}$ remains independent of $\mathbf{v}$. The discriminator, a neural network $D(\cdot)$, takes $\mathbf{v}$ as input and predicts which domain it comes from, e.g., its domain identity $k$. In this minimax game, $D(\cdot)$ attempts to classify the domain identity $k$, while the encoder inference network $q_\phi(\mathbf{v}|\mathbf{x}, \mathbf{u}, \boldsymbol{\beta})$ seeks to produce domain-invariant encodings $\mathbf{v}$ to fool the discriminator. The classification loss for this process is expressed as:

$$\mathcal{L}_{D,\phi} = \mathbb{E}_{p(k,\mathbf{x})}\mathbb{E}_{q_\phi(\mathbf{v}|\mathbf{x})}[\log D(k|\mathbf{v})] \tag{16}$$

Empirical findings generally support the assumption that $\boldsymbol{\beta}$ and $\mathbf{v}$ are independent after training, as discussed in App. C.

**Final Objective Function.** Our final objective function can be derived by combining Eqn. 11 and Eqn. 16 together:

$$\max_{\theta, \phi} \min_{D} \mathcal{L}_{DICF} = \max_{\theta, \phi} \min_{D} \mathcal{L}_{\theta, \phi} - \lambda_d \mathcal{L}_{D, \phi}$$

$$= \max_{\theta, \phi} \min_{D} \mathbb{E}_{p(\mathbf{x}, \mathbf{u}, \mathbf{R})}[\mathcal{L}_{ELBO}(\mathbf{x}, \mathbf{u}, \mathbf{R})] - \lambda_d \mathbb{E}_{p(k, \mathbf{x})}\mathbb{E}_{q_\phi(\mathbf{v}|\mathbf{x})}[\log D(k|\mathbf{v})], \tag{17}$$

with $\lambda_d$ serving as a balancing hyper-parameter.

## 3 Experiments

In this section, we demonstrate the effectiveness of our method on both model generalization and interpretability.

## 3.1 Datasets

We mainly verify our model on 3 datasets, Rec-15, Rec-30, and XMRec (Bonab et al., 2021).

**Rec-15**. We created a synthetic recommendation dataset called Rec-15, consisting of 750 users and 750 items. The items are divided into 15 domains, with each domain containing 50 items.

The intuition is that we add linearly increasing spurious features to the "true item feature". We intend for our model to infer the domain index's linear growth and learn adaptive item latent vectors across different domains. We detail below the generation processes of user features $\mathbf{u}$, item contexts $\mathbf{x}$, and ground truth ratings $R$.

For each user, we generate a 2-dimensional unit vector $[a, b]$ at random. The user feature vector $\mathbf{u}$ is then defined as $\mathbf{u} = [r + a, b]$, where $r$ is a constant offset typically set to 2.

For each item domain $k$ (where $k = 0, 1, \ldots, 14$), we compute an angle $\theta = \frac{k\pi}{30}$ and define the domain cluster center as $\boldsymbol{\mu} = [r, \ r(\cos\theta - 1), \ r\sin\theta]$. We then sample a 3-dimensional item context feature $\mathbf{x} = [c_1, c_2, c_3]$ from the normal distribution $\mathcal{N}(\boldsymbol{\mu}, \mathbf{I})$, where $\mathbf{I}$ is a 3-dimensional identity vector. The generated item feature is illustrated in Fig. 3.

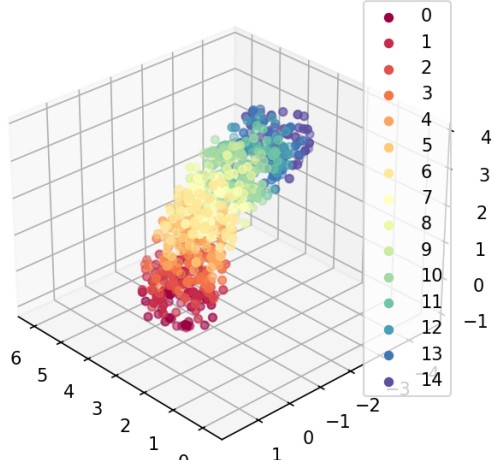

Figure 3: Visualization for the item context feature $\mathbf{x}$ of Rec-15. The color indicates the domain identities for each item feature. The item feature evolves "linearly" with the domain identity (0, 1, 2, 3, ...). This evolution represents the spurious features intentionally introduced in Rec-15. For further discussions, please refer to App. D.

The ground truth rating $R$ is generated as the dot product of the user feature $\mathbf{u} = [r + a, b]$ and the "true item feature" given by $\mathbf{v} = [c_1, \ \sqrt{(r + c_2)^2 + c_3^2} - r]$.

For training, we select items from domains 0 to 5 and use items from the remaining domains as testing data.

**Rec-30**. We also created another synthetic dataset called Rec-30 using the same procedure as Rec-15, except that it contains 30 item domains, with a total of 1,500 users and 1,500 items. Again, we select items from domains 0 to 5 for training and use the others for testing.

Table 1: Recall@300 (%) on *Rec-15* and *Rec-30*. We highlight the best result with **bold face** and the second-best results with underline.

| Dataset | PMF | CDL | DANN | MDD | TSDA | DICF (Ours) |
|---------|-----|-----|------|-----|------|-------------|
| *Rec-15* | 82.3 | 63.1 | 83.2 | 61.3 | 77.1 | **99.2** |
| *Rec-30* | 21.1 | 20.7 | 28.4 | 26.8 | 19.0 | **66.0** |

Table 2: F1-score@300 (%) on *Rec-15* and *Rec-30*. We highlight the best result with **bold face** and the second-best results with underline.

| Dataset | PMF | CDL | DANN | MDD | TSDA | DICF (Ours) |
|---------|-----|-----|------|-----|------|-------------|
| *Rec-15* | 55.0 | 50.9 | 55.3 | 49.2 | 53.7 | **60.0** |
| *Rec-30* | 32.1 | 31.8 | 40.1 | 38.5 | 29.8 | **69.8** |

**XMRec** (Bonab et al., 2021). The XMRec dataset is a cross-market recommendation dataset that encompasses 18 local markets and 16 distinct product categories, and 52.5 million user-item interactions. We utilize item descriptions from this dataset to generate item context features with the help of Sentence-BERT (Reimers & Gurevych, 2020), and subsequently create user features based on the first three items they purchased. Users with fewer than ten purchases are excluded from our experiments. To simplify the analysis, we also remove items and users that appear in multiple countries. Additionally, we exclude market data from Singapore, China, and Australia due to insufficient

numbers of items and users. After this filtering process, our experiments are conducted with 14,412 users and 48,721 items across 10 countries.

Here we focus on two tasks for XMRec:

- **Source-Rich**: Train models in data-rich source markets (Canada, France, Germany, India, US) and test in data-poor target markets (Italy, Japan, Mexico, Spain, UK).

- **Source-Poor**: Train models in data-poor source markets (Germany, India, Japan, Spain, UK) and test in data-rich target markets (Canada, France, Italy, Mexico, US).

Table 3: Recall@300 (%) for XMRec on *Souce-Rich* and *Souce-Poor* tasks across different target markets. We highlight the best result with **bold face** and the second-best results with underline.

| Task | Target Market | PMF | CDL | DANN | MDD | TSDA | DICF (Ours) |
|---|---|---|---|---|---|---|---|
| Source-Rich | Italy | 18.6 | 18.9 | 19.9 | 17.2 | 26.5 | **37.2** |
| | Japan | 65.6 | 62.0 | 68.2 | 65.8 | 66.1 | **73.0** |
| | Mexico | 9.9 | 14.5 | 11.4 | 9.6 | 10.0 | **20.6** |
| | Spain | 28.2 | 28.4 | 29.3 | 28.4 | 36.6 | **40.0** |
| | United Kingdom | 13.4 | 13.6 | 11.9 | 12.4 | 11.5 | **24.3** |
| | Average of All | 27.1 | 27.5 | 28.1 | 26.8 | 30.1 | **39.0** |
| Source-Poor | Canada | 3.8 | 5.8 | 3.8 | 4.2 | 3.4 | **7.8** |
| | France | 20.5 | 24.5 | 22.7 | 18.4 | 15.0 | **34.2** |
| | Italy | 18.4 | 18.9 | 20.5 | 19.3 | 16.4 | **35.7** |
| | Mexico | 9.1 | 14.4 | 11.2 | 10.5 | 8.2 | **15.1** |
| | United States | 1.1 | **2.0** | 1.1 | 1.0 | 1.0 | 1.3 |
| | Average of All | 10.6 | 13.1 | 11.9 | 10.7 | 8.8 | **18.8** |

## 3.2 EVALUATION METRICS

We employ two metrics for evaluation: recall@M and F1-score@M. For each user $i$, we first rank all the held-out items based on the predicted ratings. Let $J_{i,r}$ represent the $r$-th ranked item for user $i$, $S_i$ denote the set of "liked" items for user $i$, and $T$ be the total number of items ($T \geq M$). The recall@M for user $i$ is defined as follows:

$$\text{recall@M}(i) = \frac{\sum_{r=1}^{M} 1^{J_{i,r} \in S_i}}{|S_i|} \tag{18}$$

Next, we define the precision@M and F1-score@M for user $i$ as follows:

$$\text{precision@M}(i) = \frac{1}{T}\left(\sum_{r=1}^{M} 1^{J_{i,r} \in S_i} + T - M - \left(|S_i| - \sum_{r=1}^{M} 1^{J_{i,r} \in S_i}\right)\right) \tag{19}$$

$$\text{F1-score@M}(i) = \frac{2 \times \text{recall@M}(i) \times \text{precision@M}(i)}{\text{recall@M}(i) + \text{precision@M}(i)} \tag{20}$$

The final result is reported as the average for all users for both metrics.

## 3.3 BASELINES

We compare our proposed method with both the state of arts methods from both domain adaptation and cross-domain matrix factorization, including Probabilistic Matrix Factorization (**PMF**) (Mnih & Salakhutdinov, 2007), Domain Adversarial Neural Networks (**DANN**) (Ganin et al., 2016), Margin Disparity Discrepancy (**MDD**) (Zhang et al., 2019b), and Taxonomy-Structured Domain Adaptation (**TSDA**) (Liu et al., 2023). For all the domain adaptation baselines, we use the user feature as an extra input and do feature alignment on the item feature. For more implementation details, please refer to App. B.

## 3.4 RESULTS

**Rec-15 & Rec-30.** Table 1 and Table 2 show the results for the Rec-15 and Rec-30 datasets under different metrics. Our method, DICF, consistently outperforms all competing methods by a significant margin across both datasets and metrics. Specifically, for the Rec-15 dataset, DICF achieves an F1-score@300 of 60.0% and a Recall@300 of 99.2%, substantially surpassing the closest competitor, DANN, which scores 55.3% on F1 and 83.2% on Recall. On the Rec-30 dataset, DICF records an F1-score@300 of 69.8% and a Recall@300 of 66.0%, again leading over the second-best model DANN, which posts scores of 40.1% on F1 and 28.4% on Recall. These results underscore DICF's high effectiveness in identifying relevant items (high recall) and in delivering precise recommendations (high F1-score).

**XMRec.** Table 3 and Table 4 show the results for XMRec under different metrics. In the Source-Rich scenario, DICF shows excellent performance, leading both F1-score and Recall. For instance, in Italy, DICF achieves an F1-score of 51.0% and a Recall of 37.2%, surpassing the second-best model TSDA by over 10%. This high-performance trend is consistent across other countries in the source-rich scenario, helping DICF achieve the highest average score across these markets. In the Source-Poor setting, although CDL surpasses DICF in the US market, DICF still maintains a lead of over 5% on average against all other models. It is important to note that in the US, the item pool is extensive (29,390 items) and users average only about 16 rated items, which significantly impacts the performance of all models.

Table 4: F1-score@300 (%) for XMRec on *Souce-Rich* and *Souce-Poor* tasks across different target markets. We highlight the best result with **bold face** and the second-best results with underline.

| Task | Target Market | PMF | CDL | DANN | MDD | TSDA | DICF (Ours) |
|---|---|---|---|---|---|---|---|
| Source-Rich | Italy | 30.3 | 30.7 | 32.0 | 28.4 | 39.9 | **51.0** |
| | Japan | 45.0 | 43.9 | 45.7 | 45.0 | 45.1 | **46.9** |
| | Mexico | 17.9 | 24.9 | 20.1 | 17.3 | 17.9 | **33.5** |
| | Spain | 40.3 | 40.4 | 41.3 | 40.5 | 48.1 | **51.0** |
| | United Kingdom | 23.3 | 23.5 | 21.0 | 21.8 | 20.3 | **38.0** |
| | Average of All | 31.4 | 32.7 | 32.0 | 30.6 | 34.3 | **44.1** |
| Source-Poor | Canada | 7.4 | 10.9 | 7.4 | 8.0 | 6.6 | **14.5** |
| | France | 32.6 | 37.5 | 35.3 | 29.9 | 25.2 | **47.8** |
| | Italy | 30.0 | 30.7 | 32.8 | 31.2 | 27.3 | **49.5** |
| | Mexico | 16.6 | 24.9 | 20.0 | 18.8 | 15.1 | **25.9** |
| | United States | 2.1 | **3.9** | 2.1 | 2.1 | 1.9 | 2.6 |
| | Average of All | 17.7 | 21.6 | 19.5 | 18.0 | 15.2 | **28.1** |

**Visualizing Domain Indices.** We also visualize the domain indices obtained from different datasets using Principal Component Analysis (PCA).

For Rec-15 and Rec-30, as illustrated in Fig. 4, the domain indices align along a linear trajectory when plotted against the ground truth domain indices. Note that during data generation, we explicitly adding linearly growing spurious features to the "true" feature. This visualization result, combined with the numerical results presented in Table 1 and Table 2, highlights that our model successfully infers non-trivial domain indices and produces item latent vectors capable of generalizing across different domains.

For XMRec, we observe a correlation between the domain indices and the geographical/continental information of the countries in Fig. 5. For instance, in Fig. 5 (left) under the Source-Rich setting, domain indices of countries within the same continent are closer, such as the US being closer to Mexico than to India. Additionally, within the same continent, the domain index distances reflect geographical proximity, e.g., the UK is closer to France than to Spain and Italy. Under the Source-Poor setting (Fig. 5 (right)), despite some degradation in the quality of domain indices, there is still a clear clustering of European and North American countries.

Based on the analysis in Sec. 2.2, we can conclude that: **1)** During knowledge transfer, the model recognizes a closer relationship among items from countries within the same continent or with closer geographical distances. **2)** It identifies geographical/continental information as a spurious feature to

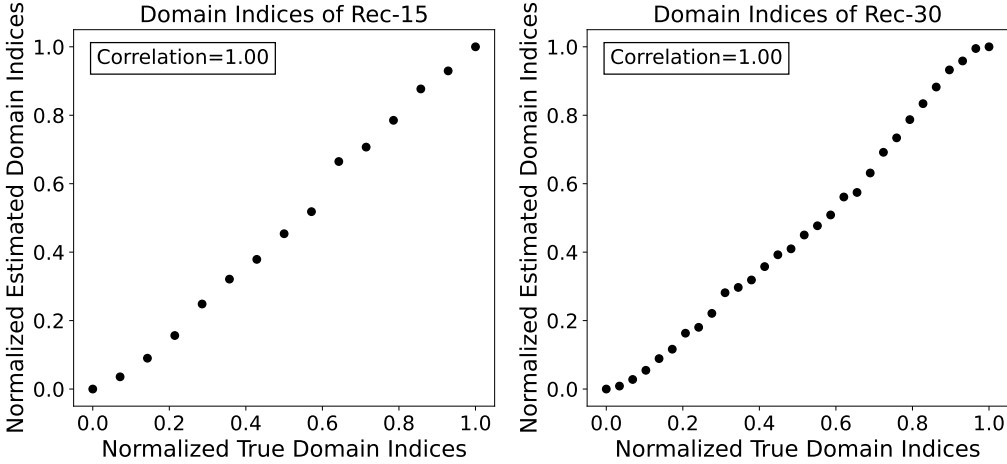

Figure 4: Normalized domain indices (reduced to 1 dimension by PCA) for 15 domains in *Rec-15* (**left**) and 30 domains in *Rec-30* (**right**). DICF successfully inferred linear domain indices, demonstrating a high correlation with the ground truth domain indices.

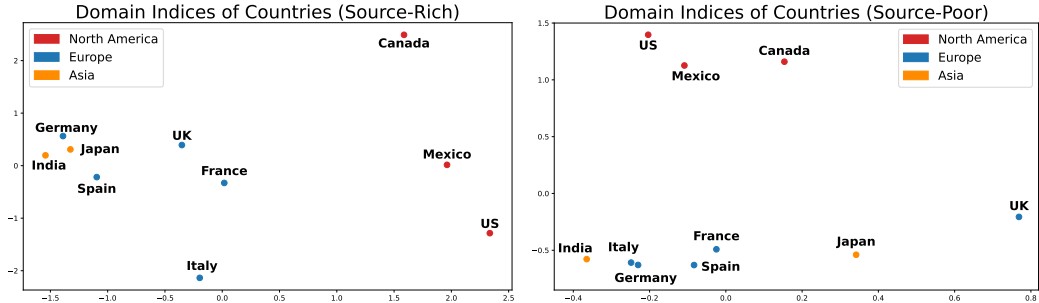

Figure 5: Inferred domain indices (reduced to 2 dimensions by PCA) for XMRec in *Source-Rich* setting (**left**) and *Source-Poor* setting (**right**). Countries are colored according to their continents. We emphasize that the model did not receive any continental/geographic information during training. See larger figures in App. A.

be removed. By eliminating the geographical/continental information from the item features, we derive a more generalizable representation suitable for cold-start item recommendations.

In summary, it is clear that our DICF framework significantly advances our understanding of the knowledge transfer process, thereby enhancing the model's interpretability.

## 4 RELATED WORKS

**Domain Adaptation.** Domain adaptation has been extensively studied (Pan & Yang, 2009; Pan et al., 2010; Ganin et al., 2016; Long et al., 2018; Saito et al., 2018; Sankaranarayanan et al., 2018; Zhang et al., 2019b; Peng et al., 2019; Chen et al., 2019; Dai et al., 2019; Nguyen-Meidine et al., 2021; Zou et al., 2018; Kumar et al., 2020; Prabhu et al., 2021; Farahani et al., 2021; Mancini et al., 2019; Tasar et al., 2020; Jin et al., 2022) to leverage prior knowledge to improve the performance of models in new environments. Various approaches have been developed for domain adaptation, with adversarial learning (Ganin et al., 2016; Ben-David et al., 2010; Tzeng et al., 2017; Zhang et al., 2019b; Kuroki et al., 2019; Chen et al., 2019; Dai et al., 2019) emerging as one of the most effective due to its high performance. Typically, adversarial learning focuses on learning domain-invariant features that generalizes across domains. Recently, several studies (Wang et al., 2020; Xu et al., 2022; Liu et al., 2023; Xu et al., 2023) have introduced domain indices alongside adversarial learning to further improve model generalization and interpretability. Building on this intuition, our work applies these ideas to the field of cross-domain recommendation.

**Cross-Domain Recommendation.** Cross-domain recommendation (Khan et al., 2017; Zhu et al., 2021a; Zang et al., 2022) focuses on utilizing information from different domains to address issues like cold-start and data sparsity. Scenarios typically differ based on whether there is overlap in users or items across domains. Our research specifically targets scenarios where there is no overlap in items and only partial overlap in users. Common approaches to these challenges include Collective Matrix Factorization (Jiang et al., 2016; Rafailidis & Crestani, 2017; Yang et al., 2017; Zhang et al., 2018; Zhu & Chen, 2022), Representation Combination for Overlapping Users (Perera & Zimmermann, 2017; Zhu et al., 2019; 2020; 2021b), and Embedding Mapping (Man et al., 2017; Wang et al., 2018; Fu et al., 2019; Li & Tuzhilin, 2021; Nahta et al., 2025). However, most of the works do not address our zero-shot problem setting, which lacks user-item interactions in the target item domains. Some methods require initial interactions (Zhang et al., 2019a; Zhu & Chen, 2022; Nahta et al., 2025), while others require additional contexts such as knowledge graphs (Bi et al., 2020; Lu et al., 2024) for making recommendations. In addition, no prior work has explicitly extracted domain indices using adversarial learning to enhance both the performance and interpretability of the model.

## 5 CONCLUSIONS

In this paper, we addressed the challenge of cold-start items in cross-domain recommendation systems by introducing the Domain Indexing Collaborative Filtering (DICF) framework. This adversarial Bayesian approach infers domain indices during the recommendation process, significantly improving performance and providing interpretability for cross-domain knowledge transfer. Future research could extend our framework to accommodate dynamic domains where user preferences and domain characteristics evolve over time. We believe that our DICF framework opens new avenues for both improving recommendation systems and advancing the interpretability of cross-domain knowledge transfer.

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

## A    LARGER FIGURES

Fig. 6 and Fig. 7 are larger figures of Fig. 5.

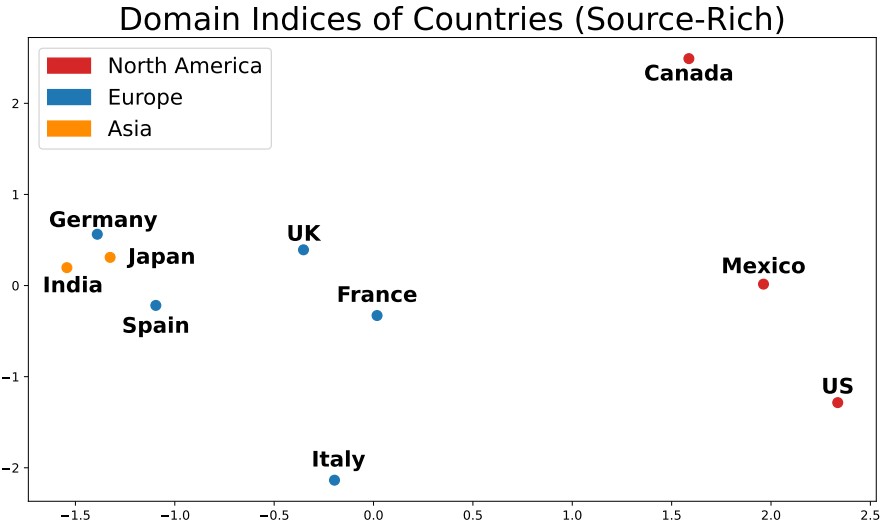

Figure 6: Inferred domain indices for XMRec in *Source-Rich* setting.

## B    IMPLEMENTATION DETAILS

**Rec-15, Rec-30.** For both datasets, We used a domain index dimension of 2. Both models were trained using the Adam optimizer, with learning rates linearly decaying from $7.88 \times 10^{-5}$ to $1 \times 10^{-8}$. For the discriminator, we used $\lambda_d = 0.32$ for Rec-15 and $\lambda_d = 5.3$ for Rec-30. A batch size of 16 was used for both models.

**XMRec.** For the source-rich task, we used a domain index of 5, and for the source-poor task, we used a domain index of 2. All models were trained using the Adam optimizer with learning rates linearly decaying from $7.88 \times 10^{-5}$ to $1 \times 10^{-8}$. For the discriminator, we used $\lambda_d = 0.7$ for

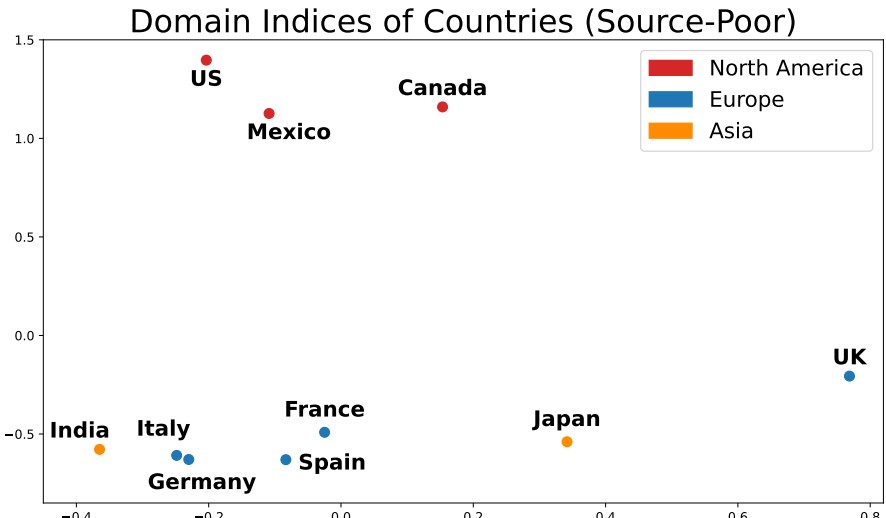

Figure 7: Inferred domain indices for XMRec in *Source-Poor* setting.

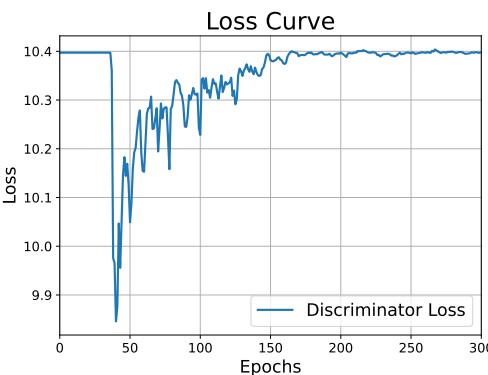

Figure 8: Visualization for discriminator loss during training. The loss initially decreases and then returns to its initial value, indicating that $D$ returns to random classification again.

the source-rich task and $\lambda_d = 0.8$ for the source-poor task. A batch size of 32 was used for both models.

All methods, including PMF, CDL, DANN, and DICF, were tuned with the same rigor. We used a latent size of 512 for all user and item latent vectors. Hyperparameters were optimized through grid search, with approximately 100 trials per model.

- **PMF**: Learning rate = 0.01, batch size = 32, and 100 training epochs.
- **DANN**: Grid search revealed that an early stopping strategy improved performance, with a learning rate = 0.01, batch size = 32, and 10 epochs across datasets.
- **CDL**: Learning rate = 1e-3, batch size = 128, and 600 training epochs due to the smaller learning rate.

All models were trained with the Adam optimizer. We ensured consistent optimization standards for DICF and all baselines.

## C  INDEPENDENCE OF DOMAIN INDICES AND DOMAIN INVARIANT FEATURE

Our method assumes that domain indices are independent of domain-invariant features after the training. Such independency assumption is generally supported by the stable pattern of discriminator

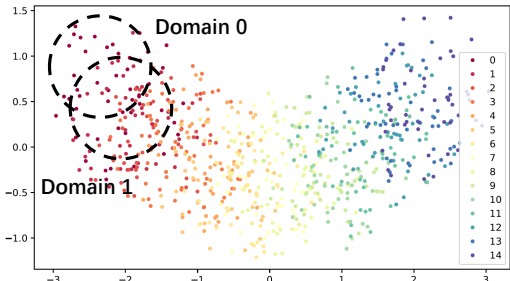

Figure 9: Visualization of the item context feature **x** of Rec-15, reduced to two dimensions using PCA. The colors represent different domain identities associated with each item feature. The item features evolve linearly across domain identities (0, 1, 2, 3, ...).

$D$'s loss. Typically, $D$ starts by randomly classifying domains. Although the loss decreases early in the training, it eventually returns to its initial value (Fig. 8), suggesting $D$ reverts to random classification due to the domain-invariant features. Xu et al., 2023 proves that if features are domain-invariant, the domain index should be independent of these features. Consistent replication of this result across various experiments and hyperparameters solidifies our confidence in this independence assumption.

## D   ITEM FEATURE VISUALIZATION FOR REC-15

To improve understanding of our datasets Rec-15, we include Fig. 9 in addition to Fig. 3, which shows a PCA transformation of the raw item features into two dimensions and uses circles to denote the first two domains. It demonstrates that the item feature evolution across domain identities (0, 1, 2, 3, ...) is linear.

