# OpenReview forum: "Domain Indexing Collaborative Filtering for Recommender System"
_ICLR.cc/2025/Conference — Submitted to ICLR 2025_

### Official Review · Reviewer_py6i · 2024-10-28

**Soundness:** 3
**Presentation:** 3
**Contribution:** 2
**Rating:** 5
**Confidence:** 2

**Summary:**

This paper focuses on cross-domain recommendation and proposes an adversarial Bayesian framework called Domain Indexing Collaborative Filtering (DICF). Extensive theoretical analysis and experiments on three datasets demonstrate the effectiveness of the approach. Furthermore, visualizing domain indices intuitively illustrates its effectiveness by showing the correlation between the domain indices and the geographical/continental information of the countries.

**Strengths:**

S1: The writing and structure of this paper is clear.

S2: The approach is innovative, employing an adversarial Bayesian method for cross-domain recommendation—a technique that has not been previously explored.

**Weaknesses:**

W1: As for the chosen baselines, since this paper focuses on cross-domain recommendation, but no baselines specific to this area were selected, instead, only baselines related to domain adaptation are used. More appropriate baselines in cross-domain recommendation are needed.

W2: Regarding the datasets, since numerous public cross-domain recommendation datasets are available, why the authors choose to use a synthetic dataset, which is relatively small. Compared to established datasets like Amazon, Rec-15, and Rec-30, which contain only 750 and 1,500 users and items, this smaller dataset is less convincing.

W3: Furthermore, the related work section should be expanded, particularly concerning cross-domain recommendation. The paper only includes studies prior to 2022, and more recent research outputs should be incorporated.

Overall, although the framework is theoretically solid, technically, it heavily relies on prior work, specifically VDI [1], which gives the impression of being incremental. Furthermore, issues remain with the selection of baselines and datasets.

[1]. Xu, Zihao, et al. "Domain-indexing variational bayes: Interpretable domain index for domain adaptation." arXiv preprint arXiv:2302.02561 (2023).

**Questions:**

See Weaknesses.

---

> ### Author Response · Authors · 2024-11-23
> **[1/2] Official Comment by Authors**
>
> Thank you for your constructive feedback. We are glad that you find our method ``"innovative"`` and our writing ``"clear"``. Below we address your questions one by one.
>
> **Q1**: As for the chosen baselines, since this paper focuses on cross-domain recommendation, but no baselines specific to this area were selected, instead, only baselines related to domain adaptation are used. More appropriate baselines in cross-domain recommendation are needed.
>
> **A**: Thank you for mentioning this. Actually, we have reviewed several studies [1, 2, 3, 4, 5], but found that they were **not applicable** for our zero-shot setting, which assumes no user-item interactions in the target item domains. Some methods require initial user-item interactions in the target domains [1, 2, 3], while others require additional contexts such as knowledge graphs [4, 5] for making recommendations. We would greatly appreciate suggestions for baselines that are adaptable to our zero-shot setting.
>
> **Q2**: Regarding the datasets, since numerous public cross-domain recommendation datasets are available, why the authors choose to use a synthetic dataset, which is relatively small. Compared to established datasets like Amazon, Rec-15, and Rec-30, which contain only 750 and 1,500 users and items, this smaller dataset is less convincing.
>
> **A**: This is a good question. The main purpose of using synthetic datasets is not to evaluate our method on large-scale data. Rather, it is to examine the robust performance of our model, given that neural networks are essentially black boxes and our method includes sophisticated adversarial training.
>
> However, we do agree that such datasets might be relatively small. This is why we also tested it on the XMRec dataset, which has **14,412 users** and **48,721 items** across **10 countries**, to confirm our model's effectiveness in real-world scenarios. In XMRec, our DICF demonstrated outstanding performance, leading by 5% across all metrics, as shown in Tables 3 and 4 of the paper.
>
> Inspired by your comments, to further validate our model, we have extended the evaluation on a new dataset, MovieLens-1M [6]. Following data cleaning, this dataset now includes information from **6,034 users** and **3,705 movies**. We segmented the dataset into 11 domains based on the year of movie release, with films from 1919-1970 and 1981-1985 designated as source domains, and films from 1971-1980 and 1986-2000 as target domains. The movie titles and descriptions were used as contextual information. We assessed the performance using precision@300 and F1-score@300. The results are shown as follows:
> | Target Movie Years | DCIF       | PMF        | DANN       | CDL        | TSDA       | MDD        |
> |--------------------|------------|------------|------------|------------|------------|------------|
> | 1971-1980          | 37.04/61.36      | 36.02/57.46| 36.98/60.95| 36.92/60.28| 37.35/62.40|**39.01/68.34**|
> | 1986-1990          | **48.02/48.77**| 44.08/41.64| 43.39/40.49| 44.76/42.78| 45.65/44.25| 35.20/28.58|
> | 1991-1995          | **46.24/57.70**| 42.75/48.90| 44.07/52.08| 43.02/49.60| 44.31/52.89| 40.42/43.44|
> | 1994-1997          | 30.71/18.90      | 24.83/14.64| 22.08/12.77| 24.74/14.58|**32.40/20.19**| 22.79/13.24|
> | 1998-2000          | **45.53/33.31**| 37.77/25.65| 39.18/26.69| 37.70/25.58| 33.88/22.20| 35.16/23.31|
> | Average              | **41.51/44.01**| 37.09/37.66| 37.14/38.60| 37.43/38.55| 38.72/40.39|34.52/35.38|
>
> The results demonstrate that our DICF outperforms most baselines in each domain and achieves improvements of at least 3.6% in the average recall and F1-score of all domains.
>
> **Q3**: Furthermore, the related work section should be expanded, particularly concerning cross-domain recommendation. The paper only includes studies prior to 2022, and more recent research outputs should be incorporated.
>
> **A**: Thank you for your comments. Following your suggestion, we have cited and expanded discussions about recent works [1,2,3,4,5] in the Cross-Domain Recommendation part of the related work section. Notably, [2,5] highlight significant works published after 2024, ensuring our discussion is up-to-date with the latest research developments.

---

> ### Author Response · Authors · 2024-11-23
> **[2/2] Official Comment by Authors**
>
> **Q4**: Overall, although the framework is theoretically solid, it heavily relies on prior work, specifically VDI [1], which gives the impression of being incremental. Furthermore, issues remain with the selection of baselines and datasets.
>
> **A**: Thank you for your comments. Our model's primary distinction from VDI lies in its application scenario. We have effectively implemented our method in the context of zero-shot cross-market recommendations. Our approach includes tailored modifications to the matrix factorization framework and the user embedding to suit this specific setting. For further details on the selection of baselines and datasets, please see our **responses to Q1 and Q2**.
>
> [1] Zhu, Yaochen, and Zhenzhong Chen. "Mutually-regularized dual collaborative variational auto-encoder for recommendation systems." Proceedings of The ACM Web Conference 2022. 2022.
>
> [2] Nahta, Ravi, et al. "CF-MGAN: Collaborative filtering with metadata-aware generative adversarial networks for top-N recommendation." Information Sciences 689 (2025): 121337.
>
> [3] Zhang, Qian, et al. "Cross-domain recommendation with semantic correlation in tagging systems." 2019 International Joint Conference on Neural Networks (IJCNN). IEEE, 2019.
>
> [4] Bi, Ye, et al. "DCDIR: A deep cross-domain recommendation system for cold start users in insurance domain." Proceedings of the 43rd international ACM SIGIR conference on research and development in information retrieval. 2020.
>
> [5] Lu, Kezhi, et al. "AMT-CDR: A Deep Adversarial Multi-channel Transfer Network for Cross-domain Recommendation." ACM Transactions on Intelligent Systems and Technology (2024).
>
> [6] Harper, F. Maxwell, and Joseph A. Konstan. "The movielens datasets: History and context." Acm transactions on interactive intelligent systems (tiis) 5.4 (2015): 1-19.

---

> > ### Comment · Reviewer_py6i · 2024-11-27
> >
> > Thanks for authors' rebuttal. Q2 and Q4 is well answered, but the answer of Q1 and Q3 is still not convincing. Since you have mentioned that this paper is based on zero-shot cross-domain recommendation, but none of related papers are cited. Zero-shot cross-domain is not a brand new problem but lots of research focus on this topic, such as [1]. The authors should conduct more extensive survey on this topic. I will retain my score.
> >
> > [1]. A Pre-trained Zero-shot Sequential Recommendation Framework via Popularity Dynamics. (RecSys '24).

---

### Official Review · Reviewer_BXJB · 2024-11-01

**Soundness:** 3
**Presentation:** 3
**Contribution:** 3
**Rating:** 5
**Confidence:** 3

**Summary:**

This paper focuses on cold-start problem in cross-domain scenarios. Two terms are intensively emphasized: domain index and spurious features (domain-specific and cannot generalize across domains). The former aggregates from spurious features and captures relationships among domains. The method is based on two properties of  spurious features. A probabilistic graphical model is used, incorporating domain-specific and invariant features into the recommendation process. The authors conducted experiments on two synthetic datasets and a real-world dataset. It enhances recommendation quality across domains by introducing interpretable domain indices.

**Strengths:**

1. The method DICF combines Bayesian modeling and adversarial learning, allowing fine-grained feature separation. This ensures that only domain-invariant features contribute to the recommendation, reducing noise and improving prediction accuracy.
2. The probabilistic graphical model and ELBO-based training enhance DICF's ability to capture complex dependencies among features and improve model generalization across different domains.

**Weaknesses:**

1. The introduction is too simple and not clear. Giving a toy example would be better to illustrate the problem of this paper.
2. This paper does not point out the challenges of problem or motivation of its method.
3. Other weaknesses can be found in Questions section.

**Questions:**

1. I think there is a mistake in Figure 2. It should be $p(\beta|\gamma)$ instead of $p(\gamma|\beta)$, right? The input should be $\gamma$ if my understanding is right.

2. Results from two synthetic datasets are not convincing because they are generated from the assumption your model has. When synthetic data is generated with assumptions aligned to the model, it can indeed inflate performance results.

---

> ### Author Response · Authors · 2024-11-23
> **[1/2] Official Comment by Authors**
>
> Thank you for your constructive feedback. We're glad you find our work effective in ``"reducing noise"``, ``"improving prediction accuracy"``, ``"capturing complex feature dependencies"``, and ``"enhancing model generalization"``. Below, we address your questions one by one:
>
> **Q1**: The introduction is too simple and not clear. Giving a toy example would be better to illustrate the problem of this paper.
>
> **A**: Thank you for your comments. Following your suggestion, we have included the following toy example in the introduction of the modified paper:
>
> Consider the tissue products from Japan, Germany, and Spain, which, despite some similarities, vary significantly in descriptions due to language differences. If a recommendation model is trained only using data from Japan and Germany (source domains), it may perform poorly in Spain (target domain) due to these linguistic discrepancies. To mitigate this, "unnecessary" or spurious features such as the language of product descriptions across domains should be removed. This resembles standardizing descriptions to English for all domains.
>
> A latent variable, termed a domain index, is learned to facilitate this removal. For example, a domain index might be an embedding that denotes the language (e.g., German, Japanese, Spanish) used in product descriptions. This index helps indicate how to eliminate spurious features and effectively serves as a ``domain embedding" that captures the essence of the domain. For instance, the domain index for German tissues might be closer to that of Spanish tissues than to Japanese tissues. Such a domain index could improve both the generalization and the interpretability of the model.
>
> Further discussion on this can be found in Sec 2.2, "Model Intuition."
>
> **Q2**: This paper does not point out the challenges of problem or motivation of its method.
>
> **A**: Our motivation is
> + to enhance the process of knowledge transfer, particularly in addressing the cold-start item issues, and
> + to increase the interpretability of the model.
>
> As previously mentioned in our **response to Q1**, one example is to enhance performance for cold-start items in Spain, using a model trained solely on data from Germany and Japan. Throughout this process, the inference of domain indices (e.g., embeddings representing the language in product descriptions) will improve our understanding of the relationships between different item domains, suggesting that German tissue products might have more similarities with Spanish ones than with Japanese ones. For more details, please refer to the first three paragraphs of our introduction.
>
> **Q3**: I think there is a mistake in Figure 2.It should be $p(\beta|\gamma)$ instead of $p(\gamma|\beta)$, right? The input should be $\gamma$ if my understanding is right.
>
> **A**: Thank you for mentioning this. There is indeed a typo. The typo is not with the text “$p(\gamma|\beta)$” inside the box, but with the text on the output arrow of the box. Specifically, on the output arrow of the box, it should be $\hat{\gamma}$, not $\hat{\beta}$. We have fixed this in the revised version of the paper.

---

> > ### Comment · Reviewer_BXJB · 2024-11-24
> > **Response to Author Rebuttal**
> >
> > Thanks to the authors for the detailed rebuttal and the efforts put into addressing the concerns. I will take this into account and update my review accordingly.

---

> ### Author Response · Authors · 2024-11-23
> **[2/2] Official Comment by Authors**
>
> **Q4**: Results from two synthetic datasets are not convincing because they are generated from the assumption your model has. When synthetic data is generated with assumptions aligned to the model, it can indeed inflate performance results.
>
> **A**: This is a good question. The main purpose of using synthetic datasets is not to evaluate our method in real-world scenarios. Rather, it is to examine the robust performance of our model, given that neural networks are essentially black boxes and our method includes sophisticated adversarial training.
>
> However, we do agree that such datasets might inflate results. This is why we also tested it on the XMRec dataset to confirm our model's effectiveness in real-world scenarios. In XMRec, our DICF demonstrated outstanding performance, leading by 5% across all metrics, as shown in Tables 3 and 4 of the paper.
>
> Inspired by your comments, to further validate our model, we have extended the evaluation on a new dataset, MovieLens-1M [1]. Following data cleaning, this dataset now includes information from 6,034 users and 3,705 movies. We segmented the dataset into 11 domains based on the year of movie release, with films from 1919-1970 and 1981-1985 designated as source domains, and films from 1971-1980 and 1986-2000 as target domains. The movie titles and descriptions were used as contextual information. We assessed the performance using precision@300 and F1-score@300. The results are shown as follows:
> | Target Movie Years | DCIF       | PMF        | DANN       | CDL        | TSDA       | MDD        |
> |--------------------|------------|------------|------------|------------|------------|------------|
> | 1971-1980          | 37.04/61.36      | 36.02/57.46| 36.98/60.95| 36.92/60.28| 37.35/62.40|**39.01/68.34**|
> | 1986-1990          | **48.02/48.77**| 44.08/41.64| 43.39/40.49| 44.76/42.78| 45.65/44.25| 35.20/28.58|
> | 1991-1995          | **46.24/57.70**| 42.75/48.90| 44.07/52.08| 43.02/49.60| 44.31/52.89| 40.42/43.44|
> | 1994-1997          | 30.71/18.90      | 24.83/14.64| 22.08/12.77| 24.74/14.58|**32.40/20.19**| 22.79/13.24|
> | 1998-2000          | **45.53/33.31**| 37.77/25.65| 39.18/26.69| 37.70/25.58| 33.88/22.20| 35.16/23.31|
> | Average              | **41.51/44.01**| 37.09/37.66| 37.14/38.60| 37.43/38.55| 38.72/40.39|34.52/35.38|
>
> The results demonstrate that our DICF outperforms most baselines in each domain and achieves improvements of at least 3.6% in the average recall and F1-score of all domains.
>
> [1] Harper, F. Maxwell, and Joseph A. Konstan. "The movielens datasets: History and context." Acm transactions on interactive intelligent systems (tiis) 5.4 (2015): 1-19.

---

### Official Review · Reviewer_VfEG · 2024-11-01

**Soundness:** 2
**Presentation:** 3
**Contribution:** 2
**Rating:** 3
**Confidence:** 3

**Summary:**

This paper addresses the cold-start challenge in cross-domain recommendation systems by introducing a novel framework, Domain Indexing Collaborative Filtering (DICF). DICF is an adversarial Bayesian approach that infers domain indices during recommendation, enhancing both performance and interpretability in cross-domain knowledge transfer.

**Strengths:**

1.	This paper presents a new adversarial Bayesian approach, DICF, which infers domain indices for cross-domain recommendation.
2.	Experimental results on both synthetic and real-world datasets demonstrate the effectiveness of DICF.
3.	The research topic, cross-domain recommendation, is promising and highly relevant to practical applications.

**Weaknesses:**

1.	The novelty of this paper is limited. The framework appears to be a combination of commonly used techniques, and the reasoning behind the selection of these techniques—such as the use of the Evidence Lower Bound—lacks clarity.
2.	The experiments are limited in scope, as they include a small number of datasets, raising concerns about the generalizability of the proposed method. Additionally, the baselines used are insufficient to verify the method’s state-of-the-art performance.
3.	Figure 3 is difficult to interpret, and it would be beneficial to consider an alternative visualization method to better convey the item context features of Rec-15.

**Questions:**

Overall, the proposed method feels formulaic and lacks a unique innovative aspect. Additionally, the experiments do not include sufficient datasets or baseline methods to fully establish the advancement of the method.

---

> ### Author Response · Authors · 2024-11-23
> **[1/2] Official Comment by Authors**
>
> Thank you for your constructive feedback. We are glad that you find our method ``"novel"`` and ``"effective"``, and our research topic ``"promising"``. Below we address your question one by one.
>
> **Q1**: The novelty of this paper is limited. The framework appears to be a combination of commonly used techniques, and the reasoning behind the selection of these techniques—such as the use of the Evidence Lower Bound—lacks clarity.
>
> **A**: Thank you for your feedback. Our key innovation is introducing a novel adversarial Bayesian framework that infers domain indices to address the cold-start problem in cross-domain recommendations. Unlike traditional methods, our approach not only **enhances performance** but also **improves interpretability** in transferring knowledge across domains. The inferred domain indices capture domain-specific features, enabling transparent and effective knowledge transfer, thereby addressing the black-box nature of prior approaches.
>
> Note that integrating matrix factorization with domain indexing to tackle the cold-start problem is **non-trivial**. Our framework leverages two main techniques: a probabilistic graphical model and adversarial learning. As per [1], the Evidence Lower Bound is necessary to infer domain indices, while the adversarial loss ensures these indices remain independent of domain-invariant features, which is critical for generalization across domains. This justifies our choice of techniques, and we have clarified this further in the appendix.
>
> **Q2**: The experiments are limited in scope, as they include a small number of datasets, raising concerns about the generalizability of the proposed method. Additionally, the baselines used are insufficient to verify the method’s state-of-the-art performance.
>
> **A**:  Thank you for your suggestion. Following your suggestion, we have extended the evaluation on a new dataset, MovieLens-1M [2]. After data cleaning, this dataset now includes information from 6,034 users and 3,705 movies. We segmented the dataset into 11 domains based on the year of movie release, with films from 1919-1970 and 1981-1985 designated as source domains, and films from 1971-1980 and 1986-2000 as target domains. The movie titles and descriptions were used as contextual information. We assessed the performance using precision@300 and F1-score@300. The results are shown as follows:
> | Target Movie Years | DCIF       | PMF        | DANN       | CDL        | TSDA       | MDD        |
> |--------------------|------------|------------|------------|------------|------------|------------|
> | 1971-1980          | 37.04/61.36      | 36.02/57.46| 36.98/60.95| 36.92/60.28| 37.35/62.40|**39.01/68.34**|
> | 1986-1990          | **48.02/48.77**| 44.08/41.64| 43.39/40.49| 44.76/42.78| 45.65/44.25| 35.20/28.58|
> | 1991-1995          | **46.24/57.70**| 42.75/48.90| 44.07/52.08| 43.02/49.60| 44.31/52.89| 40.42/43.44|
> | 1994-1997          | 30.71/18.90      | 24.83/14.64| 22.08/12.77| 24.74/14.58|**32.40/20.19**| 22.79/13.24|
> | 1998-2000          | **45.53/33.31**| 37.77/25.65| 39.18/26.69| 37.70/25.58| 33.88/22.20| 35.16/23.31|
> | Average              | **41.51/44.01**| 37.09/37.66| 37.14/38.60| 37.43/38.55| 38.72/40.39|34.52/35.38|
>
> The results demonstrate that our DICF outperforms most baselines in each domain and achieves improvements of at least 3.6% in the average recall and F1-score of all domains. For the baselines, we reviewed several studies [3, 4, 5, 6, 7], but found that they were not suitable for our zero-shot setting, which involves no user-item interactions in the target item domains. Some methods require initial user-item interactions in the target domains [3, 4, 5], while others require additional contexts such as knowledge graphs [6, 7] for making recommendations. We would greatly appreciate suggestions for baselines that are adaptable to our zero-shot setting.
>
> **Q3**: Figure 3 is difficult to interpret, and it would be beneficial to consider an alternative visualization method to better convey the item context features of Rec-15.
>
> **A**: Thank you for your feedback. In the original Figure 3, we presented the raw item feature of Rec-15, using colors to denote different domains. Our intention was to demonstrate how the item feature evolves “linearly” with the domain identity (0, 1, 2, 3, …). This evolution represents the spurious features intentionally introduced in Rec-15.
>
> Following your suggestion, to improve understanding of our datasets, we add a new figure in **Appendix D**, which shows a PCA transformation of the raw item features into two dimensions, with circles marking the first two domains. We believe the supplementary figure will help convey the item context features of Rec-15.

---

> ### Author Response · Authors · 2024-11-23
> **[2/2] Official Comment by Authors**
>
> **Q4**: Overall, the proposed method feels formulaic and lacks a unique innovative aspect. Additionally, the experiments do not include sufficient datasets or baseline methods to fully establish the advancement of the method.
>
> **A**: Please see our **responses to Q1 and Q2** for further clarification.
>
> [1]. Xu, Zihao, et al. "Domain-indexing variational bayes: Interpretable domain index for domain adaptation." arXiv preprint arXiv:2302.02561 (2023).
>
> [2] Harper, F. Maxwell, and Joseph A. Konstan. "The movielens datasets: History and context." Acm transactions on interactive intelligent systems (tiis) 5.4 (2015): 1-19.
>
> [3] Zhu, Yaochen, and Zhenzhong Chen. "Mutually-regularized dual collaborative variational auto-encoder for recommendation systems." Proceedings of The ACM Web Conference 2022. 2022.
>
> [4] Nahta, Ravi, et al. "CF-MGAN: Collaborative filtering with metadata-aware generative adversarial networks for top-N recommendation." Information Sciences 689 (2025): 121337.
>
> [5] Zhang, Qian, et al. "Cross-domain recommendation with semantic correlation in tagging systems." 2019 International Joint Conference on Neural Networks (IJCNN). IEEE, 2019.
>
> [6] Bi, Ye, et al. "DCDIR: A deep cross-domain recommendation system for cold start users in insurance domain." Proceedings of the 43rd international ACM SIGIR conference on research and development in information retrieval. 2020.
>
> [7] Lu, Kezhi, et al. "AMT-CDR: A Deep Adversarial Multi-channel Transfer Network for Cross-domain Recommendation." ACM Transactions on Intelligent Systems and Technology (2024).

---

### Official Review · Reviewer_AzFa · 2024-11-02

**Soundness:** 3
**Presentation:** 3
**Contribution:** 3
**Rating:** 6
**Confidence:** 3

**Summary:**

The paper introduces Domain Indexing Collaborative Filtering (DICF), an adversarial Bayesian framework designed to address the cold-start problem in cross-domain recommendation systems.

DICF infers domain-specific indices that capture domain relationships and enhance the interpretability of cross-domain knowledge transfer. The approach is validated on synthetic (Rec-15 and Rec-30) and real-world (XMRec) datasets, demonstrating superior performance over methods like Domain Adversarial Neural Networks (DANN) and Taxonomy-Structured Domain Adaptation (TSDA). The model is good at producing recommendations with domain-specific and generalizable features.

**Strengths:**

The paper uses a domain indexing approach with the adversarial Bayesian. This idea is innovative and enhances the interpretability of "black-box" recommendation system.

The paper addresses the cold-start issue by isolating domain-specific and domain-generalizable features, leading to better recommendations in data-poor domains. This isolation process is efficient and meaningful.

**Weaknesses:**

DICF(introduced by this paper) relies on data's meaningful domain relationships for effective knowledge transfer. This dependence may limit its generalization potential in scenarios with unrelated or highly diverse domains, where domain indices may not capture relevant transfer patterns.

The model is not explicitly designed to handle evolving domains, where domain characteristics and user preferences can change over time. This limits its utility in fast-evolving industries where product turnover and user preferences shift frequently, potentially requiring regular model retraining.

**Questions:**

Can you provide more details on the experimental setup, specifically regarding the tuning of baseline models like PMF, CDL, and DANN? Were these baselines optimized with the same rigor as DICF?

The framework assumes that domain indices remain independent of domain-invariant features. How robust is this assumption in practice, especially when there are overlapping characteristics between domains?

---

> ### Author Response · Authors · 2024-11-23
>
> Thank you for your valuable and constructive comments. We are glad that you found our work ``"innovative"`` and our isolation process ``"efficient"`` and ``"meaningful"``. Below we address your questions one by one.
>
> **Q1**: DICF(introduced by this paper) relies on data's meaningful domain relationships for effective knowledge transfer. This dependence may limit its generalization potential in scenarios with unrelated or highly diverse domains, where domain indices may not capture relevant transfer patterns.
>
> **A**: We agree that DICF requires the data domains to share some similarities for effective knowledge transfer. We would like to highlight that DICF has a key advantage on inferring such domain relationships. Specifically, DICF automatically adjusts its transfer process based on the correlation between domains. When domains are unrelated, DICF naturally relaxes the adaptation, effectively minimizing transfer and approaching a no-transfer scenario.
>
> **Q2**: The model is not explicitly designed to handle evolving domains, where domain characteristics and user preferences can change over time. This limits its utility in fast-evolving industries where product turnover and user preferences shift frequently, potentially requiring regular model retraining.
>
> **A**: This is a good question. Indeed, our current DICF model does not specifically cater to evolving data distributions, which may limit its effectiveness in rapidly changing industries. However, we would like to highlight that DICF can be naturally generalized to model time-evolving effects. This could be achieved by (1) treating “time” as an additional domain index, (2)  introducing new latent variables that account for the temporal dynamics of the data, and (3) integrating timestamps during training and inference phases.
>
> **Q3**: Can you provide more details on the experimental setup, specifically regarding the tuning of baseline models like PMF, CDL, and DANN? Were these baselines optimized with the same rigor as DICF?
>
> **A**: All methods, including PMF, CDL, DANN, and DICF, were tuned with the same rigor. We used a latent size of 512 for all user and item latent vectors. Hyperparameters were optimized through grid search, with approximately 100 trials per model.
>
> - **PMF**: Learning rate = 0.01, batch size = 32, and 100 training epochs.
> - **DANN**: Grid search revealed that an early stopping strategy improved performance, with a learning rate = 0.01, batch size = 32, and ~10 epochs across datasets.
> - **CDL**: Learning rate = 1e-3, batch size = 128, and 600 training epochs due to the smaller learning rate.
> All models were trained with the Adam optimizer. We ensured consistent optimization standards for DICF and all baselines. These details are provided in the appendix B under “Implementation Details.”
>
> **Q4**: The framework assumes that domain indices remain independent of domain-invariant features. How robust is this assumption in practice, especially when there are overlapping characteristics between domains?
>
> **A**: This assumption usually holds in practice. VDI [1] proves that if features are domain-invariant, then the domain index— a variable shared per domain—should be independent of these features.
>
> We then test this assumption by evaluating features’ domain-invariance through the discriminator $D$’s loss. Typically,
> + $D$ starts by randomly classifying domains, and
> + although the loss decreases early in the training, it eventually returns to its initial value, indicating that $D$ returns to random classification again due to the domain-invariant features (i.e., domain indices become independent of domain-invariant features).
> We consistently observe this outcome across experiments with a suitable range of hyper-parameters, *confirming that the features are indeed domain-invariant and supporting the independence assumption*. An illustrative discriminator loss curve is included in **Appendix C**.
>
> [1]. Xu, Zihao, et al. "Domain-indexing variational bayes: Interpretable domain index for domain adaptation." ICLR 2023.

---

> ### Comment · Reviewer_AzFa · 2024-11-29
>
> Thank you for addressing my comments and concerns in your rebuttal. I appreciate the detailed explanations. I believe the paper's overall contribution and impact remain consistent with my initial evaluation. Therefore, I have decided to maintain my original score.

---

### Meta-Review · Area_Chair_xwyJ · 2024-12-21

**Metareview:**

This paper proposes a method called Domain Indexing Collaborative Filtering (DICF) for cross-domain recommendations, aiming to address the recommendation challenges posed by cold-start items. The paper's motivation and idea are reasonable, and the authors conducted experiments on both synthetic and real-world datasets. However, most reviewers believe the paper currently has key weaknesses, including: 1) Insufficient baseline comparisons, particularly since reviewers mentioned existing zero-shot cross-domain recommendation methods; 2) The datasets used are not representative, and larger-scale datasets could be used for validation of the task; 3) Insufficient discussion of recent related work in cross-domain recommendation, with limited innovation in the proposed method. Considering the situation during the rebuttal phase, I think this paper is not ready for publication by far.

**Additional Comments On Reviewer Discussion:**

During the rebuttal phase, the authors addressed the issues raised by the reviewers, but they did not resolve the key concerns, especially regarding the baselines. The newly added dataset usage also lacks representativeness.

---

### Decision · Program_Chairs · 2025-01-22

Reject